# Bioaccumulation of Trace Elements in the Muscle of the Blackmouth Catshark *Galeus melastomus* from Mediterranean Waters

**DOI:** 10.3390/biology12070951

**Published:** 2023-07-03

**Authors:** Samira Gallo, Gianluca Nania, Valentina Caruso, Giorgia Zicarelli, Francesco Luigi Leonetti, Gianni Giglio, Giorgio Fedele, Chiara Romano, Massimiliano Bottaro, Olga Mangoni, Danilo Scannella, Sergio Vitale, Fabio Falsone, Giacomo Sardo, Michele Luca Geraci, Alessandra Neri, Letizia Marsili, Cecilia Mancusi, Donatella Barca, Emilio Sperone

**Affiliations:** 1Department of Biology, Ecology and Earth Sciences, University of Calabria, Via P. Bucci, 87036 Rende, Italy; samiranatural@live.it (S.G.); nania907@gmail.com (G.N.); valentina_caruso@outlook.it (V.C.); francescoluigi.leonetti@unical.it (F.L.L.); gianni.giglio@unical.it (G.G.); giorgio.fedele@unical.it (G.F.); chiara.romano@unical.it (C.R.); donatella.barca@unical.it (D.B.); 2Department of Chemical, Biological, Pharmaceutical and Environmental Sciences, University of Messina, 98166 Messina, Italy; giorgia.zicarelli@studenti.unime.it; 3Department of Integrative Marine Ecology, Genoa Marine Centre, Anton Dohrn Zoological Station, 16126 Genoa, Italy; massimiliano.bottaro@szn.it; 4Department of Biology, University of Napoli Federico II, Complesso Universitario di Monte Sant’Angelo, 80134 Naples, Italy; olga.mangoni@unina.it; 5Institute for Biological Resources and Marine Biotechnology (IRBIM), National Research Council CNR, Via Luigi Vaccara 61, 91026 Mazara del Vallo, Italy; danilo.scannella@irbim.cnr.it (D.S.); sergio.vitale@cnr.it (S.V.); fabio.falsone@irbim.cnr.it (F.F.); giacomo.sardo@irbim.cnr.it (G.S.); micheleluca.geraci2@unibo.it (M.L.G.); 6Marine Biology and Fisheries Laboratory of Fano, Department of Biological, Geological and Environmental Sciences, University of Bologna, Viale Adriatico 1/n, 61032 Fano, Italy; 7Department of Environment, Earth and Physical Sciences, Siena University, Via Mattioli 4, 53100 Siena, Italy; alessandra.neri@student.unisi.it (A.N.); letizia.marsili@unisi.it (L.M.); cecilia.mancusi@arpat.toscana.it (C.M.); 8Consorzio per il Centro Interuniversitario di Biologia Marina ed Ecologia Applicata “G. Bacci” (CIBM), Viale N. Sauro 4, 57128 Livorno, Italy; 9Environmental Protection Agency–Tuscany Region (ARPAT), Via Marradi 114, 57126 Livorno, Italy

**Keywords:** sharks, ecotoxicology, pollution, heavy metals, deep-sea

## Abstract

**Simple Summary:**

This study examines the bioaccumulation of 17 trace elements (including aluminum, arsenic, cadmium, copper, and zinc, among others) in the muscle tissue of the blackmouth catshark (*Galeus melastomus*) from various locations in the Mediterranean Sea. This research focuses provides insights into the distribution and accumulation patterns of these elements in *G. melastomus* and sheds light on the potential risks posed by chemical contamination in the Mediterranean Sea. The study underscores the significance of investigating the impacts of pollutants on marine organisms, particularly sharks, to develop effective conservation and management strategies. The data contribute to our understanding of trace element bioaccumulation in elasmobranch species, thus highlighting the importance of protecting these key ecological players in marine ecosystems.

**Abstract:**

Environmental pollution, particularly in the marine environment, has become a significant concern due to the increasing presence of pollutants and their adverse effects on ecosystems and human health. This study focuses on the bioaccumulation of trace elements in the muscle tissue of the blackmouth catshark (*Galeus melastomus*) from different areas in the Mediterranean Sea. Trace elements are of interest due to their persistence, toxicity, and potential for bioaccumulation. This research aims to assess the distribution and accumulation of trace elements in the muscle tissue of *G. melastomus* and investigate their potential impact on the deep-sea environment of the Mediterranean. The focused areas include the Ligurian Sea, the northern and central Tyrrhenian Sea, the southern Tyrrhenian Sea, the Ionian Sea, the Pantelleria Waters, and the Gela Waters. Samples were collected following established protocols, and trace element analysis was conducted using inductively coupled plasma mass spectrometry. The study provides data on the concentrations of 17 trace elements, namely aluminum, arsenic, cadmium, cobalt, copper, manganese, molybdenum, nickel, zinc, selenium, strontium, lead, chromium, iron, barium, bismuth, and uranium. The findings contribute to a better understanding of trace element bioaccumulation patterns in elasmobranch species, specifically *G. melastomus*, and highlight the potential risks associated with chemical contamination in the Mediterranean Sea. This research emphasizes the importance of studying the impacts of pollutants on marine organisms, particularly those occupying key ecological roles, like sharks, to support effective conservation and management strategies.

## 1. Introduction

Environmental pollution is an issue of particular interest to the scientific community due to the drastic increase of pollutants in the environment observed in the last decade [1]. Pollution in the marine environment is a global concern owing to the devastating effects of contaminants, which are bioaccumulated and sometimes biomagnified through trophic webs up to humans [2]. Biomagnification results in the dispersal of pollutants far from their point of origin and areas affected by intense human activities (e.g., [3,4,5]).

Trace elements are frequently assessed due to their high persistence, toxicity, and bioaccumulation affinities [6]. The World Health Organization (WHO) has divided these compounds into essential and non-essential categories: while the former play a fundamental role in biological systems (i.e., I, Zn, Se, Fe, Cu, Cr, and Mo) and could become toxic beyond a certain threshold [7], the latter do not play any physiological roles and are often toxic even at low concentrations (i.e., Hg, Pb, and Cd) [8]. Other elements (e.g., Mn, Co, As, Ni, or V) are defined by some authors as potentially essential [9].

Trace elements adsorbed onto aeolian particles and river particulates [10] are capable of being transported over long distances [11,12], and upon contact with the sea surface, some can dissolve in the upper water masses [13]. In addition, a trace element fraction, through biological processes, may reach the ocean floor as particulate matter, where they can be incorporated into sediments [14,15]. Among trace element sources in the Mediterranean Sea, aerosol powders [16] and river particulates [10] have been identified in the first 100 m of waters columns and in deep deposits (300–600 m) [14]

The tissue accumulation of trace elements depends on an organism’s water uptake. It can occur either via direct contact (e.g., through the epidermis, gills, and digestive tract) or diet [17]. Several factors influence element accumulation, such as uptake pathways, tissue function, physiological exposure, reproductive activity, life stage, and sex [18,19,20,21]; thus, this process is highly variable. Trace elements and metals that cannot be excreted are retained in the body, where they are perpetually added to over the life span of the organism [22,23]. Since marine organisms are able to accumulate trace elements, they are often considered relatively good semi-quantitative bioindicators, reflecting the occurrence and bioavailability of contaminants in the areas where they live [24].

Sharks are mostly long-lived elasmobranch species and occupy apex–meso-predator roles in their respective food webs (e.g., [25]). Therefore, they are known to accumulate high levels of metals, metalloids, and other pollutants in their tissues [26,27,28]. This feature results in deleterious effects on their normal physiology/biochemistry (e.g., immune and endocrine toxicity, reproductive deficits and tumor promotion) and ultimately on shark population dynamics [29,30,31]. Given their key role in aquatic ecosystems, especially in a closed basin such as the Mediterranean, it is crucial to study how contaminants affect these organisms [1,32].

Available information regarding the bioaccumulation of pollutants in elasmobranchs is currently limited to either specific locations of the world or to a restricted number of species [26,33,34]. Despite sharks are not being considered a popular source of fish meat [35], several shark species are targeted by fisheries to satisfy market demands for food consumption [36,37], thus raising public health concerns about the chemical contamination in elasmobranch tissues [38,39]. In particular, elasmobranchs in the Mediterranean Sea have mostly been considered low-economic bycatches and even a nuisance in the past and thus massively discarded, as in the case of the *Galeus melastomus* (Rafinesque, 1810) in the South of Sicily [40,41,42]. This small-sized shark lives in deep waters from 150 to more than 2000 m [43,44], even though it can be occasionally (especially juveniles) found over the inner shelf (50–60 m) [37,43,45,46]. *G. melastomus* feeds mainly on shelf-living species and natantian and reptantian crustaceans, as well as with teleosts and occasionally galley leftovers [47] and even birds [48]. Given the important role of the elasmobranchs in the marine ecosystem equilibrium and the paucity of data, the aim of this paper is to fill a knowledge gap by investigating and comparing the accumulation of trace elements pollutants in the muscle of the blackmouth catshark from seven areas in the Mediterranean Sea. This will contribute to assessing a potential benchmark to evaluate the impact of these elements on the Mediterranean deep-sea environment.

## 2. Materials and Methods

### 2.1. Catshark Batch and Sampling Methodology

The study area is the western–central Mediterranean Sea, which includes the northern Tyrrhenian and Ligurian Sea (geographical sub-area, GSA 9), the central and southern Tyrrhenian Sea (GSA 10), the Ionian Sea (GSA 19), and the South of Sicily (GSA 16) (Figure 1). The northern part of the Tyrrhenian Sea (GSA 9) is more heterogeneous from a ecological and morphological point of view, and the continental shelf extends up to 150 m and to a maximum depth of between 2000 and 2200 m. The southern–central Tyrrhenian Sea (GSA 10) is characterized by two principal abyssal plains with a maximum depth between 2900 and 3600 m and by the presence of two important submarine volcanos. The principal currents in the GSA 9 originate from the wind, and they are subjected to significant seasonal variations, while the geophysical, morphological, and dynamic structures of the GSA 10 are one of the most complex of all the Italian peninsulas [49,50]. The Ionian Sea (GSA 19) is the deepest sea of the entire Mediterranean Basin with a mean depth of ca. 2000 m. The Taranto Valley divides the Ionian Sea into the western and the south-eastern slopes [51] The Strait of Messina marks the boundary between the Tyrrhenian and the Ionian Seas, where strong vertical gradients, horizontal gradients, and tidal currents originate [52]. The Strait of Sicily (GSA 16) is an area of transition between the sub-basins of the western and eastern Mediterranean and includes the Sicilian and Tunisian platforms. The geomorphology is very complex, characterized by several sea mountains (banks) composed of sedimentary or volcanic rocks, such as the volcanic island of Pantelleria located between the Sicilian and Tunisian platforms. This topography affects the currents around the banks, resulting in significant upwelling, which, in turn, increases overall productivity, making this channel one of the most important biodiversity hotspots in the Mediterranean basin [53].

The specimens of the GSA9 and GSA16 were collected during the international bottom trawl survey in the Mediterranean (MEDITS), while the samples of the GSA10 and GSA19 were collected via commercial bottom trawling. In Table 1, geographical coordinates, sampling period, number of specimens, and catching depth are reported. Specimens of *G. melastomus* were collected following the Italian law “Decreto Legislativo 4 marzo 2014, n. 26”, which implemented the “European Directive 2010/63/UE” and is recognized as using the guidelines put forward in Ebert et al. [54].

### 2.2. Tissue Dissection

The specimens—previously stored at −20 °C—were thawed at room temperature for 24 h prior for further manipulation [55]. Each sample was photographed, and its sex and biometric measurements were recorded: fresh weight (W), total length (TL), and fork length (FL). W was taken to the nearest 0.01 g whereas TL to the 0.1 cm. Subsequently, the specimens were dissected using surgical stainless steel straight scissors with alternating tips, precision stainless steel tweezers, and disposable surgical scalpels (measures n.10 and n.11). A cut was made ventrally, from the cloacal opening up to the cardiac cavity. Muscle biopsies were taken from both sides, below the peritoneal membrane, up to about 1 gr each side, and the samples were stored in dry conditions (−20 °C) in 1.5 mL Eppendorf tubes until analysis.

### 2.3. Analysis of Trace Elements

The analysis provided the concentrations, expressed in ppb (parts per billion), of 17 trace elements: aluminum (Al), arsenic (As), cadmium (Cd), cobalt (Co), copper (Cu), manganese (Mn), molybdenum (Mo), nickel (Ni), zinc (Zn), selenium (Se), strontium (Sr), lead (Pb), chromium (Cr), iron (Fe), barium (Ba), bismuth (Bi), and uranium (U). This analysis was conducted using an inductively coupled plasma mass spectrometer (ICP-MS, PerkinElmer, ElanDRCe model). The DRC consists of a quadrupole placed inside an enclosed reaction chamber. This quadrupole eliminates polyatomic interferences caused by the combination of plasma gases and sample matrix constituents before they can enter the analyzing quadrupole. This ion-molecule chemistry uses a reaction gas (CH_4_) to “chemically scrub” polyatomic or isobaric species from the ion beam before they enter the analyzer, resulting in improved detection limits for traditionally difficult elements, including As, Cr, Fe, Se, and others. The plasma torch, however, is powered using Argon with a purity of 5.00. The strength of this equipment is its fast multiplexing, allowing the acquisition of the data of several elements at once with high precision.

Prior to analysis, in order to solubilize the samples, muscle tissue samples were chemically digested, according to the protocol of De Donato et al. [56], and is described as follows: (a) Drying—the tissue was air dried in a laminar flow hood (Herasafe, model KS-12 class II) for 24 h. (b) Weighing—each sample was weighed in its Teflon (TFM) digestion vessel on an analytical balance with the sensitivity of 0.0001 (g) (Sartorius, model CP324S-0CE). (c) Acid digestion—12 mL of ultrapure HNO_3_ (64–69%) was added to the tissue, the vessels (hermetically sealed) were installed on a carousel and placed inside an EMC microwave oven (Mars 6). A program that provides a rising temperature ramp up to 180 °C was then applied for 20 min, followed by 20 min of cooling. After cooling, the vessels were worn out. (d) Evaporation—the content was transferred from the digestion vessel into larger vessels, taking care to rinse any residue with ultra-pure water (UPW). These vessels were then placed on a hot plate at 200 °C for about 2–3 h (t changes according to the starting volume) in order to eliminate acid fumes and to avoid damage to the instrumentation used for analysis. (e) Make up to the mark—the concentrated solution obtained by evaporation was transferred into glass matrasses and diluted to the volume of 100 mL using ultra-pure water. The solution was then homogenized inside the matrasses by shaking (manual stirring) in order to avoid the uneven distribution of the elements at the bottom of the flask. (f) Storage—the solution was temporarily stored in sterile containers at 4 °C until analysis.

During phases (a) and (b), the muscles’ water percentage was also measured, thus allowing the conversion of the dry weight values, returned by the ICP-MS, to wet weight (mg/Kg), using the equation of Gaion et al. [57]:WW = DW × (1 − WP)
where WW is the wet weight, DW is the dry weight, and WP is the water percentage loss during the drying stage. The blank was prepared with ultra-pure water and ultrapure nitric acid (100 µL HNO_3_:10 mL UPW). The internal standard (Re) was added to both blank, standard, and other solutions. The ICP-MS was calibrated using the ICP multi-element standard solution VI for ICP-MS (Merck). The accuracy of the method was evaluated using the Lobster Hepatopancreas (Tort-3; National Research Council Canada) as the certified reference material (CRM). The same preparation procedure was used on the CRM and these solutions were used as unknown samples during the analytical sequence. During the analysis, the blank and the CRM were processed by the ICP-MS at the beginning and at the end and were also interspersed with every six samples (checking that the values were always in the norm and that the drift of the instrument did not affect the results obtained). Accuracies were better than 8%, with most elements within ±5%. The detection limit was set at three times the standard deviation of the blanks and its value oscillated around 0.2 mg/Kg.

### 2.4. Data Analysis

For the comparison of trace elements mean values between sexes and among areas, the Mann–Whitney U-test and Kruskal–Wallis test were used, respectively, with Dunn test as the post hoc test. To investigate a possible correlation between trace element concentration and the growth of the species, the bioaccumulation of each area was compared to the TL of the specimens by means of linear regression. All statistical analyses were carried out using the Instat 3.0 program. We considered five significance levels: no significance (NS, *p* > 0.05), not quite significant (§, 0.05 ≤ *p* ≤ 0.01), significant (*, 0.01 ≤ *p* ≤ 0.001), very significant (**, 0.001 ≤ *p* < 0.0001), and extremely significant (***, *p* ≤ 0.0001). In addition, the trace element pollution index (TEPI) was calculated as stated by Richir and Gobert [58]:TEPI = (Cf_1_ × Cf_2_ … Cf_n_)^1/n^
where Cf_n_ is the normalized mean concentration of the trace element—calculated as the mean concentration of each TE per location divided by the overall mean of that TE—and n is the total number of examined potentially toxic elements [59]. Once the TEPI values have been calculated, they can be classified qualitatively according to a three-level contamination scale, defined by the quartile method [59].

## 3. Results

The batch of samples was composed of 78 specimens with a total of 38 females and 40 males, including 68 animals in the adult size class (TL > 34 cm) and 10 in the sub-adult size class (23 cm > TL > 34 cm), according to Ebert et al. [54]. The mean values of TL, W, and sample size are reported in Table 2.

The results obtained for 17 trace elements were grouped by sex and areas. The average values of DW concentrations (mg/Kg) were summarized in Table 3.

On the basis of the average concentration values per element, males tended to show higher values than females; however, the Mann–Whitney U-test shows a statistically significant difference only for As (*, *p* < 0.01) (Figure 2). The specimens originating from the CT and the LS showed greater bioaccumulation—six elements each—while the specimens from PW showed higher bioaccumulation values of Al. The samples originating from GW waters showed the lowest bioaccumulation—12 elements. In more detail, Figure 3 highlights the number of elements of which the different areas showed higher concentrations when compared to the other sites.

As for the comparison of the trace elements between different areas (Table 4), specimens from the Pantelleria Waters (PW) showed a statistically significant difference in five of the elements (Al, As, Cd, Co, and Pb; *p* < 0.0001), while the specimens from Gela Waters (GW) showed a statistically significant difference in six elements (As, Cd, Co, Cu, Pb, and Zn; *p* < 0.0001). Moving to the Tyrrhenian Sea areas (GSA 9 and 10), specimens from ST showed no statistically significant differences, whereas those caught in CT, NT, and LS areas displayed statistically significant differences in five (Al, Fe, Mn, Se, and Sr; 0.03 < *p* < 0.0001), three (Bi, Sr, and U; *p* < 0.0001), and six (Al, Cr, Mo, Ni, Se and Sr; 0.02 < *p* < 0.0001) elements, respectively. Finally, the specimens originating from the GSA 19 (IS) differed in two elements (Ba, *p* < 0.0001, and Fe, *p* < 0.03).

Regarding the relationship between trace elements and the growth of the species, As was the only significant element in every area, with a positive trend (i.e., the concentrations for this element increases with increasing TL), while five elements (Cd, Cr, Cu, Mo, and Zn) showed a negative trend or no trends across all sites. Specimens from CT showed the greatest statistical significance—in nine elements—and the greatest number of negative trends. In general, despite slight differences between areas, more negative correlations were observed. Statistical results are summarized in Figure 4.

Scales of abundance for each sex, sampling areas, and overall amounts (Table 5) are obtained using concentration mean values.

Taking into account the most abundant elements found (As, Fe, Al, Zn, Sr, Cr, Cu, Mn, and Ni) and the two elements within the Regulations EC No 1881/2006 (Cd and Pb), we calculated the mean concentrations for the WW and determined the percentages of contribution for each area (Figure 5). The 100% refers to the total amount of each element obtained, so that the concentration per site can be observed in percentage and the comparison between locations is visible.

The water percentage in muscles was found to be 70.45% (average), then the average values of the concentrations for the WW were obtained.

TEPI was calculated and reported in Figure 6. The first and third quartiles were used as thresholds to classify the level of contamination impact and they were equal to 0.563 and 0.997, respectively. The sites PW and GW were found to be the least affected by trace element pollution (TEPI < first quartile) while the sites CT and LS were the most impacted (TEPI > third quartile). The areas ST, IS, NT, and LS stood in the middle of the contamination scale (first quartile > TEPI< third quartile).

## 4. Discussion

Although previous studies on bioaccumulation for *G. melastomus* in the Mediterranean Sea are indeed present in the literature, e.g., [57,60,61], our study is, to the best of our knowledge, the first to have compared many trace elements (17) across different sites (7).

Males showed a trend for higher mean values for the concentration of trace elements than females, supporting previously published findings that described a lower bioaccumulation in females of many species.

Energy investment during reproduction, the size at which they reach sexual maturity, the energy cost of gamete synthesis, and the transfer of contaminants during oocyte development, have all been suggested as putative explanations for the observed sexual dimorphic variation [57,62].

Interestingly, females showed a very high bioaccumulation of As, which is one of the most controversial elements. According to the WHO, As is classified as potentially essential. Its presence in the water can originate from both natural and anthropogenic sources, and numerous studies investigating human toxicity are underway. Storelli et al. [63] had already reported a positive correlation between body size and the bioaccumulation of As, further confirmed by our results.

Both sexes showed an identical bioaccumulation of Pb with no significant differences. This non-essential element can be toxic to marine organisms as well as humans, even at low concentrations [8]. The accumulation begins at the embryonic stage [64]—if the egg is exposed to this pollutant—and it continues through subsequent developmental stages via ingestion (mainly through prey [65]). However, it is an elements that can be expelled from the body throughout an animal’s life [66,67].

In terms of areas, CT and LS specimens showed the highest bioaccumulation (35.3% each), followed by IS specimens, then NT, ST and finally PW. These differences could be explained by the hydrodynamic conditions that characterize the various areas, as observed in other contexts [68,69]. Nevertheless, the excellent difference in bioaccumulation rates in PW and GW compared to all the other areas stands out in particular. Two reasons could be proposed to justify this result. First of all, both large and mesoscale oceanographic patterns in the Sicilian Channel are strongly related to the combining effects of wind and seafloor topography; in particular, wind-induced coastal upwelling along the southern Sicilian coast could impact the pollution dispersion toward the east [70,71,72]. The removal of pollutants towards the east could therefore limit their bioaccumulation, especially in demersal species. Secondly, they could simply be more remote areas and therefore subject to low levels of pollution.

The correlation between trace elements and growth has showed an overall negative trend, probably due to a change in diet (in cases where food was the main source), a possible “moving away” dispersal from the source or to the mechanisms of metabolization and the expulsion of pollutants that developed during growth (e.g., development of oocytes and deposition). It is worth noting that while samples from most sites do not have a wide variety of lengths (TL > 34 cm), only the central Tyrrhenian Sea batch included sub-adults (23.6 < TL < 51.0 cm). The results of the present study showed values for Cd and Pb equal to 0.023 mg/Kg (WW) and 0.029 mg/Kg (WW), respectively. Both values exhibited below maximum levels for certain contaminants in foodstuffs (Cdlimit = 0.050 mg/Kg; Pblimit = 0.30 mg/Kg) set by the “Commission Regulation (EC) No 1881/2006 of 19th December 2006”, suggesting that the levels of this trace element in blackmouth catshark in the western–central Mediterranean Sea are safe for human consumption.

By comparing the As content in the blackmouth catshark with previous studies, we found that the mean value at our sites (2020–2021) is 27.004 mg/Kg WW (91.383 mg/Kg DW), as opposed to the quantified maximum value in the Aegean Sea of 13.07 mg/Kg WW reported in Storelli and Marcotrigiano [60] in other Mediterranean areas (two in the Adriatic Sea, one in the Ionian Sea, and one in the Aegean Sea).

Focusing only on the northern Tyrrhenian Sea, the elements As, Cd, Pb, and Cu appeared in amounts of 121.07 mg/Kg, 0.14 mg/Kg, 0.16 mg/Kg, and 1.75 mg/Kg, respectively. Previously, Gaion et al. [57] found greater values for As and Pb (174.5 mg/Kg and 0.35 mg/Kg) and almost equal or slightly lower values for Cd and Cu (0.1 mg/Kg and 1.7 mg/Kg).

Mille et al. [61] analyzed two different populations of *G. melastomus*—one in the Mediterranean Sea (Gulf of Lions) and one in the Atlantic Ocean (Bay of Biscay). The comparison between the mean values of their populations (Atlantic and Mediterranean) and our seven areas from the western–central Mediterranean Sea showed higher concentrations for the specimens of the present study, i.e., Cd, Pb, Zn, and Ni, while the average of our values for Cu was lower than the average they obtained in the Mediterranean Sea.

Furthermore, the trace elements pollution index (TEPI) for the western–central Mediterranean Sea (0.873) estimated in the present study yielded results that were lower than those reported by Mille et al. [61] in the northern–western Mediterranean Sea (1.13). Instead, the lower values obtained in the GSA 16 (PW = 0.563 and GW = 0.488) were similar to those observed in the northern–eastern Atlantic Ocean (0.7) and in the Bay of Biscay (0.56).

## 5. Conclusions

In conclusion, these results enrich our knowledge about this deep-water species and provide more insights into the condition of the Mediterranean Sea’s environment in terms of pollution by trace elements. This study, nevertheless, represents a starting point for further studies, aimed at deepening our understanding of what occurs in different tissues and what the consequences for human health could be.

## Figures and Tables

**Figure 1 biology-12-00951-f001:**
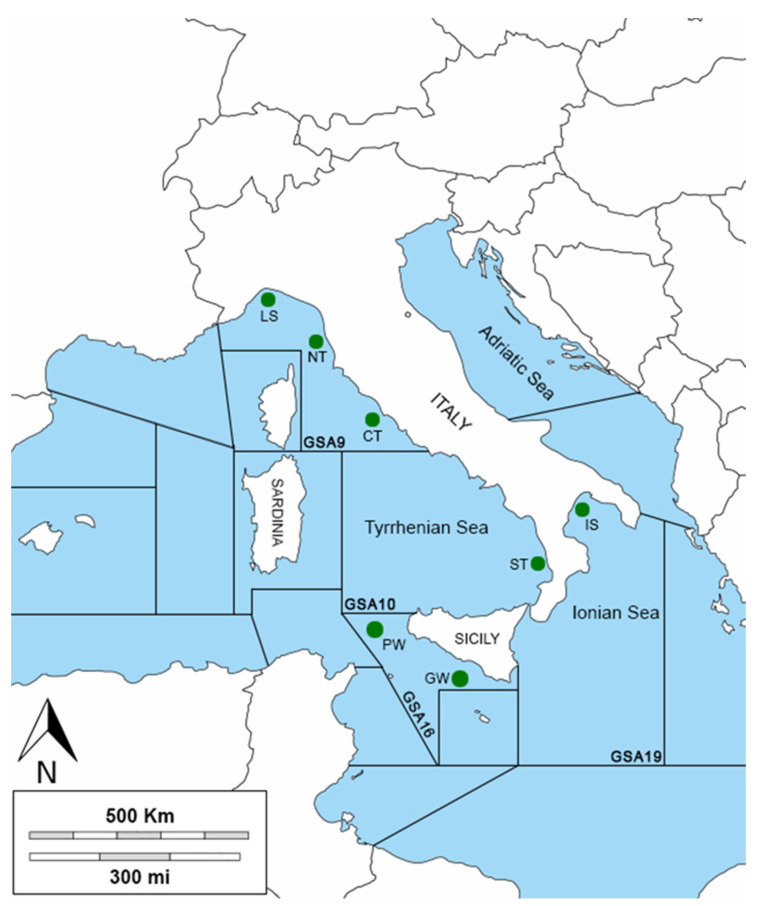
Sampling area: The green circles indicate the sampling localities. Three areas are located in the GSA9 (Ligurian Sea, LS; northern Tyrrhenian Sea, NT; and central Tyrrhenian Sea, CT), one area in the GSA10 (southern Tyrrhenian Sea, ST), two areas in the GSA16 (Pantelleria Waters, PW, and Gela Waters, GW), and one area in the GSA19 (Ionian Sea, IS).

**Figure 2 biology-12-00951-f002:**
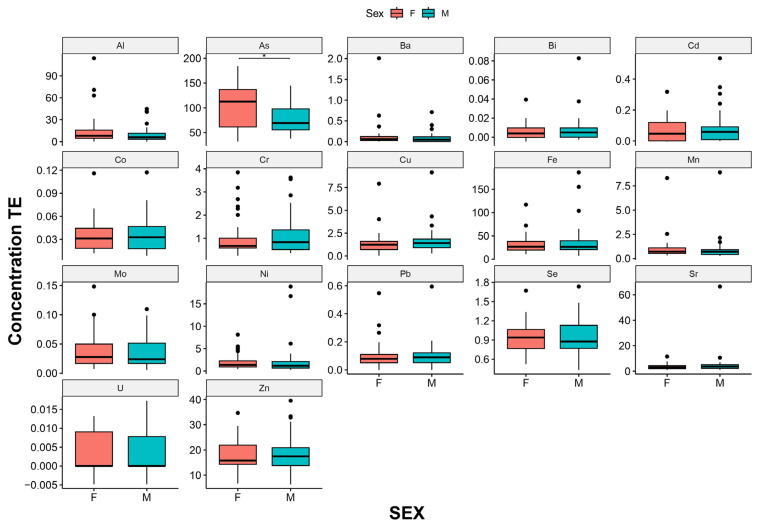
Comparison of TE concentrations between sexes.

**Figure 3 biology-12-00951-f003:**
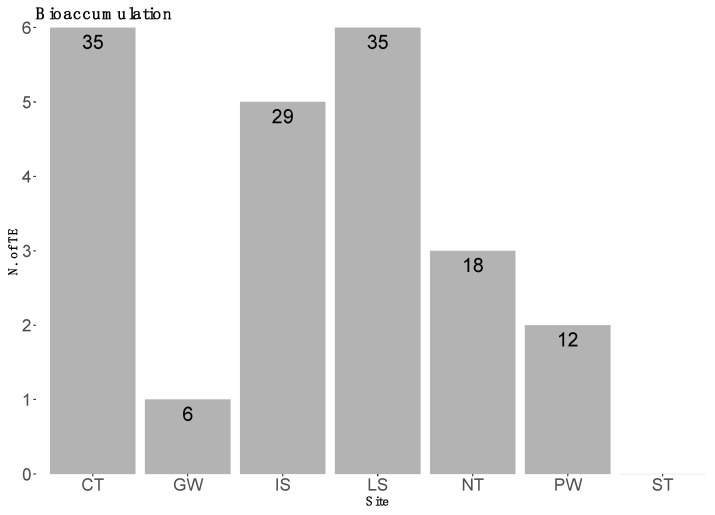
Percentage of elements for which the sampling areas showed greater bioaccumulation values.

**Figure 4 biology-12-00951-f004:**
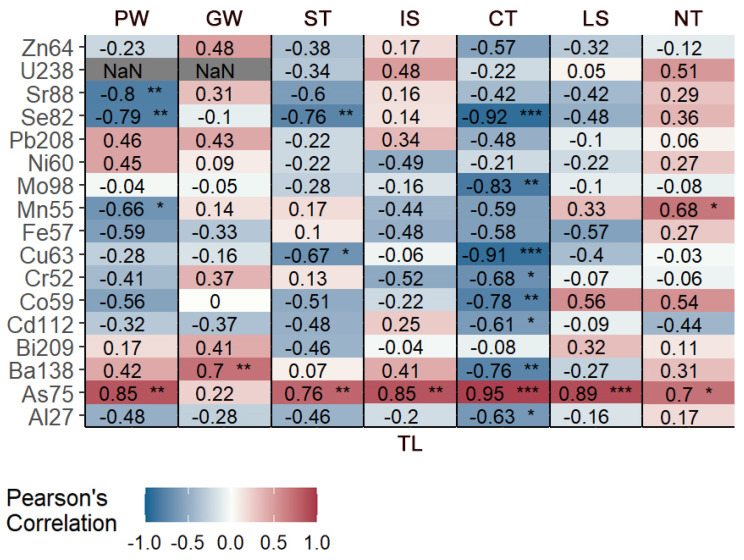
Statistical significance of the correlation between elements and TL for each sampling area (the comparison is not performed between areas). We considered three significance levels: *, significant (0.01 ≤ *p* ≤ 0.001), **, very significant (0.001 ≤ *p* < 0.0001), and ***, extremely significant (*p* ≤ 0.0001).

**Figure 5 biology-12-00951-f005:**
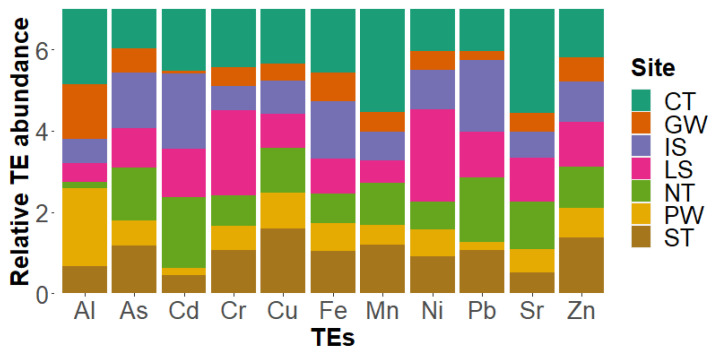
Abundance contribution of each site for specific TE. The concentration values were normalized to the means in similar manner to TEPI calculations. Only values for eleven elements (Al, As, Cd, Cr, Cu, Fe, Mn, Ni, Pb, Sr, and Zn) are shown in the figure.

**Figure 6 biology-12-00951-f006:**
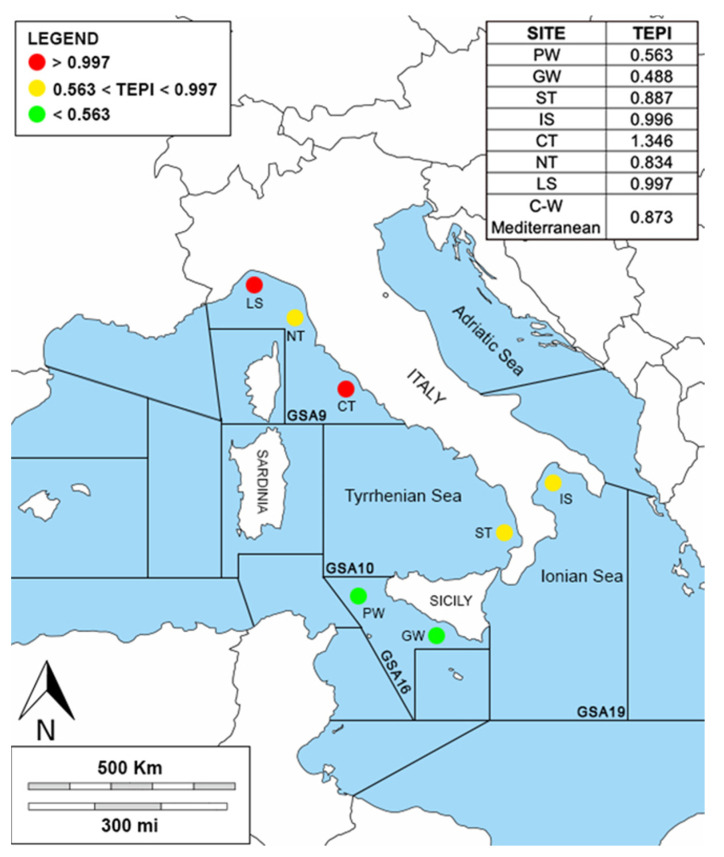
Visual representation of the three-contamination scale in the study area, where the legend shows the values under the first quartile (green), the values between the first and the third quartile (yellow), and the values over the third quartile (red). At the top right: TEPI results for each site.

**Table 1 biology-12-00951-t001:** This table outlines the sampling areas in detail, with coordinates, sampling period, number of specimens sampled, and the analysis and catching depth (in m).

Site	Latitude °N	Longitude °E	Sampling Period	N° Specimens	Depth (m)
Pantelleria Waters (PW; GSA16)	37.4374	11.2093	November2021	3	543
	37.5395	11.2947	November 2021	3	519
	37.5385	11.3775	November 2021	1	390
	37.4669	11.3356	November 2021	1	585
	37.3384	11.4340	November 2021	1	421
	37.4895	11.5687	November 2021	1	584
Gela Waters (GW; GSA16)	36.5458	13.5052	October 2021	12	587
	36.5183	13.5554	October 2021	2	648
Southern Tyrrhenian Sea (ST; GSA10)	38.9410	16.1253	November 2020	4	40
	38.9410	16.1253	February 2021	6	40
	38.9410	16.1253	April 2021	1	40
Ionian Sea (IS; GSA19)	40.0418	16.8842	February 2021	7	300
	40.0418	16.8842	August 2021	2	300
Central Tyrrhenian Sea (CT; GSA9)	41.2758	12.1098	October 2020	6	476
	42.0477	11.0979	October 2020	5	347
(Northern Tyrrhenian Sea (NT; GSA9)	43.3531	9.3552	October 2020	11	634
Ligurian Sea (LS; GSA9)	43.4845	9.3827	October 2020	2	412
	44.0668	9.3205	October 2020	4	392
	44.1574	8.4084	October 2020	5	582
	43.5852	8.1727	October 2020	1	471

**Table 2 biology-12-00951-t002:** Composition of the samples’ batch: TL mean and standard deviation (Sd), W mean and Sd, and number of specimens of each sex and for each area.

Site	Sex	Total Length (cm)			Weight (g)		
		Mean	Sd	N	Mean	Sd	N
CT	F	40.40	11.72	6	198.67	132.03	6
	M	33.80	9.18	5	106.20	82.65	5
GW	F	41.96	5.46	5	210.60	76.68	5
	M	39.88	3.16	9	171.33	46.52	9
IS	F	53.68	0.80	6	419.67	64.55	6
	M	46.83	1.21	3	262.33	21.36	3
LS	F	46.70	6.17	6	253.00	81.55	6
	M	36.27	1.15	6	108.33	33.29	6
NT	F	48.85	0.92	2	291.00	12.73	2
	M	44.96	1.35	9	210.89	14.00	9
PW	F	34.04	8.22	7	137.14	99.68	7
	M	42.10	3.82	3	216.67	20.03	3
ST	F	49.02	2.06	6	355.00	28.40	6
	M	41.54	6.36	5	171.00	65.43	5

**Table 3 biology-12-00951-t003:** Mean values of the concentrations obtained and Sd for each TE, divided into sex and site.

Site	Sex	Al			As			Ba			Bi			Cd			Co		
		Mean	Sd	N	Mean	Sd	N	Mean	Sd	N	Mean	Sd	N	Mean	Sd	N	Mean	Sd	N
CT	F	24.9604	43.8582	6	104.4854	45.7561	6	0.1731	0.2277	6	0.0018	0.0018	6	0.1091	0.0555	6	0.0561	0.0319	6
	M	19.6982	21.0614	5	73.2153	32.6417	5	0.2712	0.2876	5	0.0030	0.0042	5	0.1429	0.1321	5	0.0731	0.0269	5
GW	F	16.4685	9.5237	5	62.7695	12.9673	5	0.0712	0.0646	5	0.0036	0.0016	5	0.0000	0.0000	5	0.0148	0.0029	5
	M	16.0513	10.5739	9	51.1478	10.3954	9	0.0224	0.0273	9	0.0033	0.0029	9	0.0093	0.0217	9	0.0144	0.0034	9
IS	F	6.6891	3.3471	6	148.3959	22.9589	6	0.4543	0.7704	6	0.0130	0.0049	6	0.1643	0.0865	6	0.0293	0.0104	6
	M	7.9501	3.0680	3	91.7474	30.5497	3	0.0881	0.0337	3	0.0119	0.0071	3	0.1247	0.1011	3	0.0373	0.0315	3
LS	F	5.0471	6.2344	6	113.3381	31.8111	6	0.0956	0.0639	6	0.0082	0.0040	6	0.1038	0.0726	6	0.0362	0.0102	6
	M	6.2727	4.1885	6	65.8161	7.2587	6	0.1414	0.0639	6	0.0100	0.0001	6	0.0915	0.0554	6	0.0316	0.0078	6
NT	F	3.9014	4.2061	2	139.5839	42.1486	2	0.0894	0.0920	2	0.0235	0.0224	2	0.0710	0.0416	2	0.0626	0.0106	2
	M	1.4861	2.2023	9	116.9547	17.0229	9	0.0570	0.0638	9	0.0169	0.0276	9	0.1589	0.1615	9	0.0373	0.0087	9
PW	F	30.2927	26.0157	7	52.8619	19.6956	7	0.0239	0.0287	7	0.0041	0.0037	7	0.0193	0.0388	7	0.0168	0.0056	7
	M	6.1830	3.3549	3	69.9069	3.1306	3	0.0000	0.0000	3	0.0007	0.0009	3	0.0000	0.0000	3	0.0092	0.0002	3
ST	F	6.4905	2.9157	6	137.6272	19.5088	6	0.0273	0.0278	6	−0.0017	0.0015	6	0.0299	0.0198	6	0.0464	0.0094	6
	M	10.0209	8.3239	5	76.5792	14.0002	5	0.0997	0.1206	5	0.0012	0.0058	5	0.0455	0.0131	5	0.0643	0.0160	5
**Site**	**Sex**	**Cr**			**Cu**			**Fe**			**Mn**			**Mo**			**Ni**		
		**Mean**	**Sd**	**N**	**Mean**	**Sd**	**N**	**Mean**	**Sd**	**N**	**Mean**	**Sd**	**N**	**Mean**	**Sd**	**N**	**Mean**	**Sd**	**N**
CT	F	1.2467	0.4414	6	2.0306	1.0834	6	44.5548	36.0351	6	2.3719	2.9102	6	0.0273	0.0229	6	2.7728	3.0889	6
	M	1.9226	1.0278	5	2.2480	0.8876	5	65.9720	51.8599	5	2.8295	3.4053	5	0.0399	0.0256	5	1.5357	0.9876	5
GW	F	0.5732	0.0853	5	0.5652	0.3272	5	23.8598	20.1310	5	0.5558	0.1655	5	0.0165	0.0081	5	0.8274	0.3731	5
	M	0.4580	0.0864	9	0.7065	0.4221	9	25.7014	15.2076	9	0.4640	0.1260	9	0.0173	0.0050	9	1.1572	1.1164	9
IS	F	0.5447	0.0958	6	1.2644	0.8154	6	33.1030	12.8473	6	0.6035	0.1913	6	0.0558	0.0260	6	1.5340	0.7224	6
	M	0.8413	0.4601	3	1.3098	0.3654	3	80.1899	91.9531	3	0.9421	0.6575	3	0.0677	0.0369	3	3.1071	2.7841	3
LS	F	2.4347	1.2185	6	1.0708	0.2832	6	28.4636	7.5858	6	0.6426	0.0869	6	0.0890	0.0381	6	2.6529	1.7739	6
	M	2.0862	0.7985	6	1.5119	0.7352	6	32.0847	6.0149	6	0.4716	0.2188	6	0.0863	0.0107	6	7.1409	8.3576	6
NT	F	0.7113	0.0428	2	1.9775	0.2748	2	30.0820	7.4380	2	2.0744	0.6559	2	0.0185	0.0027	2	3.0952	2.9957	2
	M	0.8125	0.1274	9	1.6983	0.4223	9	23.7441	4.6300	9	0.8258	0.0835	9	0.0194	0.0075	9	1.0547	0.5625	9
PW	F	0.7152	0.1670	7	1.7017	2.7474	7	28.6477	13.6299	7	0.5583	0.1464	7	0.0224	0.0115	7	1.7712	1.7079	7
	M	0.4913	0.1061	3	0.5748	0.3558	3	13.5406	8.7026	3	0.3686	0.1157	3	0.0114	0.0058	3	0.7264	0.1532	3
ST	F	0.8133	0.2600	6	1.5124	0.2283	6	29.6092	21.7559	6	1.1494	0.2635	6	0.0297	0.0116	6	1.7598	0.6320	6
	M	1.5640	1.1181	5	3.7194	3.2494	5	44.2360	37.1268	5	1.2723	0.5001	5	0.0427	0.0127	5	2.1432	0.7332	5
**Site**	**Sex**	**Pb**			**Se**			**Sr**			**U**			**Zn**					
		**Mean**	**Sd**	**N**	**Mean**	**Sd**	**N**	**Mean**	**Sd**	**N**	**Mean**	**Sd**	**N**	**Mean**	**Sd**	**N**			
CT	F	0.1141	0.0810	6	1.1218	0.3217	6	5.7811	2.9863	6	−0.0023	0.0021	6	19.6623	4.8288	6			
	M	0.0962	0.0560	5	1.2839	0.3914	5	17.8381	27.3701	5	−0.0016	0.0022	5	23.9009	9.1823	5			
GW	F	0.0240	0.0303	5	0.7023	0.1359	5	2.1478	1.3294	5	0.0000	0.0000	5	10.8729	5.5265	5			
	M	0.0208	0.0296	9	0.6826	0.1287	9	2.0619	0.6231	9	0.0000	0.0000	9	10.4669	3.5376	9			
IS	F	0.2158	0.1825	6	1.0689	0.1379	6	2.8573	0.9431	6	0.0066	0.0051	6	18.3294	4.0012	6			
	M	0.1077	0.0131	3	1.0204	0.1996	3	2.6596	0.2479	3	0.0030	0.0053	3	17.0362	3.8732	3			
LS	F	0.1037	0.0489	6	1.1218	0.1127	6	4.2370	1.9062	6	0.0049	0.0054	6	17.5750	4.5390	6			
	M	0.1244	0.0487	6	1.2493	0.0959	6	5.2720	1.0225	6	0.0016	0.0040	6	21.8926	5.0578	6			
NT	F	0.1256	0.0340	2	1.1282	0.1624	2	5.2536	1.3427	2	0.0128	0.0006	2	15.1830	1.2399	2			
	M	0.1724	0.1630	9	0.9017	0.1002	9	5.1453	0.9122	9	0.0092	0.0063	9	18.5478	5.6382	9			
PW	F	0.0193	0.0256	7	0.7985	0.1721	7	2.9651	1.5824	7	0.0000	0.0000	7	15.6048	6.7693	7			
	M	0.0190	0.0330	3	0.6592	0.1154	3	1.3321	0.3298	3	0.0000	0.0000	3	7.9719	0.8732	3			
ST	F	0.1080	0.0428	6	0.7746	0.0588	6	2.0465	0.5173	6	0.0056	0.0039	6	24.3586	6.6737	6			
	M	0.1091	0.0494	5	0.9298	0.1920	5	2.5676	1.5937	5	0.0086	0.0027	5	24.6685	5.7404	5			

**Table 4 biology-12-00951-t004:** Statistical significance of trace elements within each sampling area. We considered three significance levels: NS, not significant (*p* > 0.05); §, not quite significant (0.05 ≤ *p* ≤ 0.01), and ***, extremely significant (*p* ≤ 0.0001). Furthermore, “+” represents grater bioaccumulation rates and “−” represents lower bioaccumulation rates.

	PW	GW	ST	IS	CT	NT	LS
Al	*** +	NS	NS	NS	*** +	NS −	*** +
As	*** −	*** −	NS	NS +	NS	NS	NS
Ba	NS −	NS	NS	*** +	NS	NS	NS
Bi	NS	NS	NS −	NS	NS	*** +	NS
Cd	***−	*** −	NS	NS +	NS	NS	NS
Co	*** −	*** −	NS	NS	NS +	NS	NS
Cr	NS	NS −	NS	NS	NS	NS	*** +
Cu	NS	*** −	NS +	NS	NS	NS	NS
Fe	NS	NS	NS	§ +	§ +	NS	NS
Mn	NS −	NS −	NS	NS	*** +	NS	NS
Mo	NS	NS −	NS	NS	NS	NS −	*** +
Ni	NS	NS −	NS	NS	NS	NS	§ +
Pb	*** −	*** −	NS	NS +	NS	NS	NS
Se	NS	NS −	NS	NS	*** +	NS	*** +
Sr	NS	NS −	NS	NS	*** +	*** +	*** +
U	undetected	undetected	NS	NS	undetected	*** +	NS −
Zn	NS	*** −	NS +	NS	NS	NS	NS

**Table 5 biology-12-00951-t005:** Scales of abundance for sex, sampling areas, and overall amounts.

GSA 16 (Pantelleria Waters)	*As* > *Fe* > *Al* > *Zn* > *Sr* > *Ni* > *Cu* > *Se* > *Cr* > *Mn* > *Mo* = *Pb* > *Ba* > *Co* > *Cd* > *Bi* > *U*
GSA 16 (Gela waters)	*As* > *Fe* > *Al* > *Zn* > *Sr* > *Ni* > *Se* > *Cu* > *Cr* > *Mn* > *Ba* > *Pb* > *Mo* > *Co* > *Cd* > *Bi* > *U*
GSA 10 (Southern Tyrrhenian Sea)	*As* > *Fe* > *Zn* > *Al* > *Cu* > *Sr* > *Ni* > *Mn* > *Cr* > *Se* > *Pb* > *Ba* > *Co* > *Cd* = *Mo* > *U* > *Bi*
GSA 19 (Ionian Sea)	*As* > *Fe* > *Zn* > *Al* > *Sr* > *Ni* > *Cu* > *Se* > *Mn* > *Cr* > *Ba* > *Pb* > *Cd* > *Mo* > *Co* > *Bi* = *U*
GSA 9 (Central Tyrrhenian Sea)	*As* > *Fe* > *Al* > *Zn* > *Sr* > *Mn* > *Ni* > *Cu* > *Cr* > *Se* > *Ba* > *Cd* > *Pb* > *Co* > *Mo* > *Bi* = *U*
GSA 9 (Northern Tyrrhenian Sea)	*As* > *Fe* > *Zn* > *Sr* > *Al* > *Cu* > *Ni* > *Mn* > *Se* > *Cr* > *Pb* > *Cd* > *Ba* > *Co* > *Bi* = *Mo* > *U*
GSA 9 (Ligurian Sea)	*As* > *Fe* > *Zn* > *Al* > *Ni* > *Sr* > *Cr* > *Cu* > *Se* > *Mn* > *Ba* > *Pb* > *Cd* > *Mo* > *Co* > *Bi* > *U*
Males (all populations)	*As* > *Fe* > *Zn* > *Al* > *Sr* > *Ni* > *Cu* > *Cr* > *Mn* > *Se* > *Pb* > *Ba* > *Cd* > *Co* = *Mo* > *Bi* > *U*
Females (all populations)	*As* > *Fe* > *Zn* > *Al* > *Sr* > *Ni* > *Cu* > *Cr* = *Mn* > *Se* > *Ba* > *Pb* > *Cd* > *Mo* > *Co* > *Bi* > *U*
Total (all populations)	*As* > *Fe* > *Zn* > *Al* > *Sr* > *Ni* > *Cu* > *Cr* > *Mn* > *Se* > *Ba* > *Pb* > *Cd* > *Co* = *Mo* > *Bi* > *U*

## Data Availability

Data are available on request due to restrictions, e.g., privacy or ethical.

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
