# Peer review of "Bioaccumulation of Trace Elements in the Muscle of the Blackmouth Catshark Galeus melastomus from Mediterranean Waters"

_biology, 2023, doi:10.3390/biology12070951_

Round 1
Reviewer 1 Report
General comment:
This study examined the bioaccumulation of trace elements in Galeus melastomus muscle tissue from different locations in the Mediterranean. Comparisons were made on 17 trace elements in different growth stages and genders of catshark, revealing the accumulation patterns of these trace elements. This will raise awareness of food safety and environmental pollution. This study is meaningful for the Mediterranean.
Specific comments:
Introduction
Line 74 It is easy to misinterpret the author's statement, which leads the reader to believe that the only non-essential metals are Hg, Pb and Cd.
Conclusions
Line 353 What does “-” mean?
References
Please confirm that the author name in references 7 and 75 is correct.
Supplementary Materials
In Table S2 and S3, it might be better to add the standard deviation.
Appropriate revisions need to be made to the manuscript.
Author Response
Responses to Reviewer 1
Point 1. This study examined the bioaccumulation of trace elements in Galeus melastomus muscle tissue from different locations in the Mediterranean. Comparisons were made on 17 trace elements in different growth stages and genders of catshark, revealing the accumulation patterns of these trace elements. This will raise awareness of food safety and environmental pollution. This study is meaningful for the Mediterranean.
Response 1: Thank you so much for your comments.
Point 2. Specific comments:
Introduction
Line 74 It is easy to misinterpret the author's statement, which leads the reader to believe that the only non-essential metals are Hg, Pb and Cd.
Response 2: According to the World Health Organization (WHO) only Hg, Pb and Cd are classified as non-essential trace elements. All other elements, which are neither essential nor non-essential, are classified as potentially essential (Guéguen et al., 2011).
Point 3. Conclusions
Line 353 What does “-” mean?
Response 3: It was an error, thank you for noticing it. That was edited.
Point 4. References
Please confirm that the author name in references 7 and 75 is correct.
Response 4: We remove some references under the ask of another reviewer so now the bibliography should be correct.
Point 5. Supplementary Materials
In Table S2 and S3, it might be better to add the standard deviation.
Response 5. This was edited accordingly.
Reviewer 2 Report
The paper shows trace element concentrations in Black-mouth catshark of the Mediterranean, as clearly explained by the authors these studies are of importance since very few studies have been made and there is little information on TE concentrations on these organisms, hence the relevance of the study. It shows several TE concentrations on different sites, which is important since comparisons can be made.
Overall, I think it is well written, but I feel a lack of connection between the introduction, the results and the discussion. The introduction describes very well pollution issues and trace element presence, as well as the treatment of the catshark by catch. I think the lack of connection comes from some statements made in the introduction, such as when it says it is crucial to study how contaminants affect these organisms in line 9, since no experiments or histological studies were done, there is no way to say how are they affected, I think it should focus on the concentrations that would be found in the organisms and that the study sites.
In the regards it also seems important to me to show some characteristics of the Mediterranean Sea, maybe describe in this point some of the currents or some of the possible pollutants that could be influencing these sites, as discussed later in the discussion section.
Then in the results section concentrations of females and males are not shown but in the supplementary material, however, is the first part of the discussion section, it would seem important to me to show such results in a table in the actual manuscript and not in the supplementary section. The results highlight a lot of differences in TE across sites, that is then discuss, that is why I think more description of the sites is needed.
In the discussion section some maximum levels permitted for food are shown for two TE, if the other TE do not have limitations perhaps add a little paragraph explaining as to why limits have not been set.
The last paragraph seems to me that should not be the last statement of your discussion, something else should be said, maybe that catshark is a good indicator? Or that some more studies are needed or that the TEPI clearly reflects the dynamics of the system that is then very summarized in the conclusion?
In material and methods some things are missing:
How much of the sample was taken or weighed?
I do not understand the methodology used to determine bioaccumulation, do you mean the trace element concentration? Accumulated in regards to what? Maybe I was lost in this part of the methodology, but could you please clarify
And also in this sense, how percentage of bioaccumulation was obtained, I guess 100% corresponds to all of the trave element concentrations, but I think it should be clear from the methodology section.
Letters on figure 2 are very small and hard to read, as well as the figure 4 where the color site is described.
I am not adding any file.
Author Response
Responses to Reviewer 2
Point 1: The paper shows trace element concentrations in Black-mouth catshark of the Mediterranean, as clearly explained by the authors these studies are of importance since very few studies have been made and there is little information on TE concentrations on these organisms, hence the relevance of the study. It shows several TE concentrations on different sites, which is important since comparisons can be made.
Overall, I think it is well written, but I feel a lack of connection between the introduction, the results and the discussion. The introduction describes very well pollution issues and trace element presence, as well as the treatment of the catshark by catch. I think the lack of connection comes from some statements made in the introduction, such as when it says it is crucial to study how contaminants affect these organisms in line 9, since no experiments or histological studies were done, there is no way to say how are they affected, I think it should focus on the concentrations that would be found in the organisms and that the study sites.
In the regards it also seems important to me to show some characteristics of the Mediterranean Sea, maybe describe in this point some of the currents or some of the possible pollutants that could be influencing these sites, as discussed later in the discussion section.
Then in the results section concentrations of females and males are not shown but in the supplementary material, however, is the first part of the discussion section, it would seem important to me to show such results in a table in the actual manuscript and not in the supplementary section. The results highlight a lot of differences in TE across sites, that is then discuss, that is why I think more description of the sites is needed.
Response 1: Thank you for your consideration and your revisions.
Concerning the introduction and the apparent disconnection between the issues addressed and the study conducted, we wanted to justify the choice of providing detailed information also on the consequences of exposure to trace element pollution for ichthyofauna, in order to provide a state of the art as complete as possible but in the final part of the introduction, we specify that the study represents an assessment of a benchmark to evaluate future impacts.
Regarding the Mediterranean Sea and the sites, we edited the text accordingly with your suggestions adding more details in material and method section; we also moved the tables from supplementary materials to the actual manuscript.
Point 2. In the discussion section some maximum levels permitted for food are shown for two TE, if the other TE do not have limitations perhaps add a little paragraph explaining as to why limits have not been set.
Response 2: In the discussions we specify that the comparison is made on the basis of the current legislation and not arbitrarily. Having stressed in the introduction the classification of trace elements (essential, non-essential and potentially essential), we do not think it is necessary to reiterate the reason why the values of essential elements (e.g. Zn, Fe, Cu) are not regulated or, in any case, it is not the subject of this study to go into the substance of the regulations. The Commission Regulation (EC) No 1881/2006 defines the toxicity thresholds in food only for non-essential elements, namely Cadmium, Lead and Mercury (the latter not considered as not present in our study).
Point 3. The last paragraph seems to me that should not be the last statement of your discussion, something else should be said, maybe that catshark is a good indicator? Or that some more studies are needed or that the TEPI clearly reflects the dynamics of the system that is then very summarized in the conclusion?
Response 3: Many thanks for this suggestion, we added a last statement.
Point 4. In material and methods some things are missing:
How much of the sample was taken or weighed?
Response 4: If "sample" refers to the tissue sample, the method does not provide a defined amount of fresh weight because it is essential only that the dry weight actually used during the acid attack is 0.1 gr. We have, however, integrated into the manuscript (L152) how much fresh weight was taken.
Point 5. I do not understand the methodology used to determine bioaccumulation, do you mean the trace element concentration? Accumulated in regards to what? Maybe I was lost in this part of the methodology, but could you please clarify
Response 5: By bioaccumulation we mean the concentration of trace elements in the considered tissue and it is an absolute value.
Point 6. And also in this sense, how percentage of bioaccumulation was obtained, I guess 100% corresponds to all of the trave element concentrations, but I think it should be clear from the methodology section.
Response 6: If the comment refers to the results summarized in Figures 2 and 4, the percentages refer respectively to the number of TEs (in relation to the total of elements analyzed) for which each site showed higher concentrations than the others, and the total of ppm per element obtained during the analysis (so as to quantify the "contribution" of each site in achieving that total). This calculation does not provide any specific method or analysis, but to avoid misunderstandings we have on your suggestion specified in the manuscript to what the percentages refer.
Point 7. Letters on figure 2 are very small and hard to read, as well as the figure 4 where the color site is described.
Response 7: This was edited accordingly.
Reviewer 3 Report
In the manuscript “Bioaccumulation of Trace Elements in the muscle of the blackmouth catshark Galeus melastomus Rafinesque, 1810 from Mediterranean waters”, concentrations of 17 elements are determined in blackmouth catshark samples from 7 regions around Italy and then subjected to statistical treatment.
The addressed topic is important (element levels/contamination in marine environments). The results are not particularly exciting or novel, but it is good to have such data for monitoring, future comparison, and baseline establishment. However, the results need to be presented and discussed in a completely different way to warrant publication. Concerning the language, most parts of the manuscript are easy to read and follow. Overall, I suggest publication after major revision.
Major points:
Results:
All results are hidden in the Supplementary, and most of the tables/figures in the main manuscript are poorly selected and do not present any relevant information to the reader. Comparisons are made between male and female and for the 7 different locations, and for each element, differences are identified. However, this is not true at all. For example, lines 209ff: “Females showed a higher bioaccumulation in 7 elements (…) while males showed higher bioaccumulation in 10 elements (…).” Even though there were only mean values (no standard deviation) given in the supplementary, one can easily see that this statement is not true. In some cases, the concentrations were completely the same for males and females. Furthermore, the differences between males and females is one of the first things mentioned in the discussion, and therefore it would be important to actually undermine this claim with actual data in the main manuscript.
The situation is similar for the comparison of the different locations. Again, standard deviations are missing, but the “greater bioaccumulation” is often not there. What can be seen, however, is the similarity of locations that are close together, e.g. PW and GW, and I am surprised that this was not mentioned at all. In this context, Figure 2 should be removed completely.
Very late in the manuscript (L 334ff), the results from one sampling area are compared to literature. It is claimed that the literature values are “greater” or “lower” for certain elements, but the differences are sometimes not really there. Take for example Cu, where the literature value of 1.7 mg/kg is regarded as “slightly lower” than the manuscript’s 1.75 mg/kg. So far, I can only see a different number of significant digits, but no difference in magnitude. Again, standard deviations would be essential here.
As can be seen by the previous concerns, the complete lack of standard deviations (or similar) is a big problem, because such information is essential to get an idea about the variability of the data.
Table 3 is not necessary. It makes no sense to compare the levels of different elements to each other, because they naturally come in different levels.
Figure 4 also makes no sense. Why should the element concentrations of the 7 locations add up to 100%? These 7 locations were just selected by the authors (no explanations given) and do not form a closed environment or anything similar.
Figure 4 and Figure 5 (wrongly named “Figure 4” as well) and the related parts in the main text: I am very skeptical about the calculated “TEPI” values. First of all, what is the “normalized mean concentration” (L197)? And second, I do not see why this calculation warrants the identification of two of the locations as “contaminated”. The values just happen to be the two highest ones of the 7 tested locations. But the concentrations can still be very well in a normal, unpolluted range.
It is stated that As is bioaccumulated by in the shark tissues (L 289ff). Just a few lines later, a reference is mentioned where no bioaccumulation was found, so the opposite of the manuscript’s findings. Yet, there is no discussion about this, where this controversy could come from.
So, please completely revise the presentation and discussion of your results. It is good to compare males and females, and the different locations. But this does not mean that there always has to be a “higher” or “lower”. Sometimes, there just is no difference. But this is also a valid finding. When comparing the locations with each other, it could be worth to also state how big the found differences are, so if one location should be regarded as polluted (and if so, then where does this pollution come from?), or if the values are all in an acceptable range. From what I could see from the data so far, it seems that the PW and GW values are always very similar to each other, and often lower than the others. Why? Could this have to do with their position, which seems a bit further away from coasts? I believe that some actually relevant, small discussions can be written from this dataset, to make the reader more interested in the study.
To replace the rather irrelevant figures and tables (as mentioned above), I would love to see actual concentrations in a table. For example, Supplementary Table S3 could be moved to the main manuscript (+standard deviations). Further, the results for males vs. females could be presented as bar charts (+standard deviations) in the main manuscript.
Quality control:
I highly appreciate that a reference material is mentioned. It would be important that other information on quality assurance/quality control is given. For example: Were blanks measured? Drift controlled? What kind of calibration was done? How was the ICPMS operated (e.g. selected m/z values and gas modes, sensitivity)?
Sample preparation:
Why were the samples evaporated after digestion? What is meant by “glass matrasses”? How did you ensure that there was no significant contribution of the glass containers to the element concentrations? (Normal glass is very “dirty” when it comes to certain trace elements!) Why was “stirring” of the diluted samples necessary, and what was used for that purpose?
Official limits & co.: In L 290f, it is stated that the WHO classifies As as potentially essential. If such controversial claims are maid, it is crucial to give the appropriate references. Further, in L326f, the As results are compared to a “maximum limit of 15 mg/Kg”, which simply does not exist for foodstuff. Probably, the one cited is for feed, but this has no relevance here.
References: For a rather short paper, there are a lot of references (81!). I suggest reducing the number of references, even though I appreciate the thoroughness. Several times, the DOI is not correct (e.g.: ref. 21 or 26, but also others). In other references, information is missing, such as the publication year (e.g. ref 7 or 12, but also others). Please check all references for correct and complete information.
------------------------------
Smaller comments:
Title: Remove “Rafinesque, 1810” to make the title more concise and easier to understand. Such detailed information on the investigated organism is important, but belongs to the introduction and the Materials & Methods sections.
L 81: Change “(Martin et al., 1989)” to a proper reference.
Materials and methods: The description of the sampling areas are confusing, because only the “larger” areas are specifically mentioned in the main text, and the individual 7 points only in Figure 1 and Table 1.
Table 1: Make sure that the text in the first row has enough space between the different columns.
Table 1: Since the sharks probably swim a bit up and down all the time, I don’t think it makes sense to give depths up to 0.5 m precision.
L 141: the “r” in “room temperature” should be lower case.
L 151: Change “[…]for17[…]” to “[…]for 17[…]”
L 155: Change “[…] the inductively coupled […]” to “[…] an inductively coupled […]”
L 159: The two references explain exactly the same procedure. Remove one of them.
L 222-224: This is the original text of the template. Remove.
L 279: add a closing “)” after the references.
Supplementary: Make sure that the table captions are on the same page as the actual tables.
Author Response
Responses to Reviewer 3
Point 1. In the manuscript “Bioaccumulation of Trace Elements in the muscle of the blackmouth catshark Galeus melastomus Rafinesque, 1810 from Mediterranean waters”, concentrations of 17 elements are determined in blackmouth catshark samples from 7 regions around Italy and then subjected to statistical treatment.
The addressed topic is important (element levels/contamination in marine environments). The results are not particularly exciting or novel, but it is good to have such data for monitoring, future comparison, and baseline establishment. However, the results need to be presented and discussed in a completely different way to warrant publication. Concerning the language, most parts of the manuscript are easy to read and follow. Overall, I suggest publication after major revision.
Response 1: Thank you so much for your comments and revisions.
Major points:
Point 2. Results:
All results are hidden in the Supplementary, and most of the tables/figures in the main manuscript are poorly selected and do not present any relevant information to the reader. Comparisons are made between male and female and for the 7 different locations, and for each element, differences are identified. However, this is not true at all. For example, lines 209ff: “Females showed a higher bioaccumulation in 7 elements (…) while males showed higher bioaccumulation in 10 elements (…).” Even though there were only mean values (no standard deviation) given in the supplementary, one can easily see that this statement is not true. In some cases, the concentrations were completely the same for males and females. Furthermore, the differences between males and females is one of the first things mentioned in the discussion, and therefore it would be important to actually undermine this claim with actual data in the main manuscript.
Response 2: We decide to move Supplementary materials in the manuscript so that the results can be easily read and checked by the reader. Concerning the comparison between sexes, we agree with the point that in some cases the concentrations were the same between males and females, but the general trend is that males showed higher values even if these are not statistically significant. We edit that part in order to generate less confusion.
Point 3. The situation is similar for the comparison of the different locations. Again, standard deviations are missing, but the “greater bioaccumulation” is often not there. What can be seen, however, is the similarity of locations that are close together, e.g. PW and GW, and I am surprised that this was not mentioned at all. In this context, Figure 2 should be removed completely.
Response 3: The standard deviation has been added as required and the comparison between sites – in terms of the number of elements for which each site has independently shown higher concentrations (summarized in Figure 2) – seems to us however an interesting information to return. But if you continue to feel that it is superfluous, it won’t be a problem to remove it.
Point 4. Very late in the manuscript (L 334ff), the results from one sampling area are compared to literature. It is claimed that the literature values are “greater” or “lower” for certain elements, but the differences are sometimes not really there. Take for example Cu, where the literature value of 1.7 mg/kg is regarded as “slightly lower” than the manuscript’s 1.75 mg/kg. So far, I can only see a different number of significant digits, but no difference in magnitude. Again, standard deviations would be essential here.
As can be seen by the previous concerns, the complete lack of standard deviations (or similar) is a big problem, because such information is essential to get an idea about the variability of the data.
Response 4: That was edited accordingly.
Point 5. Table 3 is not necessary. It makes no sense to compare the levels of different elements to each other, because they naturally come in different levels.
Response 5: Always taking into account that the elements are naturally present in the environment with orders of magnitude different from each other, we think it might be interesting to note the slight differences between sexes and populations observable from the scales of abundance (for example, Cu that only in the ST site is among the first 5 elements for abundance or Pb that - even if slightly - is more abundant in males than in females). For this reason, we would prefer to keep the table but are willing to remove it if you still consider it superfluous.
Point 6. Figure 4 also makes no sense. Why should the element concentrations of the 7 locations add up to 100%? These 7 locations were just selected by the authors (no explanations given) and do not form a closed environment or anything similar.
Response 6: Although we agree with the reviewer’s comment, the aim of Fig 4 was to show “graphically” the contribution of each TE in relation to the different “sites”. That is a normalization of the level detected for each TE across the sites. Thus, it is logical that the TE concentration adds up to 100%.
The "100%" in Figure 4 refers to the total ppm observed per element. The purpose of Figure 4 is to highlight graphically how much each sampling site participates in the total obtained per element, in order to further emphasize the difference between geographical locations per element.
If the figure is analyzed vertically - by column/TE - it is immediate to notice which site has returned higher or lower concentrations for that given element, while if the figure is observed horizontally - for sites/locations - differences in concentrations between elements are visible.
The “more abundant TEs” were selected based on the 50th percentile method, i.e. we considered abundant each TE which overall mean concentration was above the 50th percentile.
The 7 locations were selected based on availability of samples.
Point 7. Figure 4 and Figure 5 (wrongly named “Figure 4” as well) and the related parts in the main text: I am very skeptical about the calculated “TEPI” values. First of all, what is the “normalized mean concentration” (L197)? And second, I do not see why this calculation warrants the identification of two of the locations as “contaminated”. The values just happen to be the two highest ones of the 7 tested locations. But the concentrations can still be very well in a normal, unpolluted range.
Response 7: The normalized mean is the normalization of the concentration of each TE across different locations, that is the mean concentration of each TE per location divided by the overall mean of that TE. While the TEPI cannot provide hard cut offs for demining whether a site is contaminated or not, it offers an indication. The higher the index value is, the more contaminated the monitored station is.
In addition, it allows the comparison among different studies, which might have used different methodologies and/or different Ns (both in terms of TE detected and sites). The actual cutoffs are calculated based on the quartile methods.
Point 8. It is stated that As is bioaccumulated by in the shark tissues (L 289ff). Just a few lines later, a reference is mentioned where no bioaccumulation was found, so the opposite of the manuscript’s findings. Yet, there is no discussion about this, where this controversy could come from.
Response 8: In that paragraph we talk about bioaccumulation, understood as the concentration of As detected in the muscle tissue of sharks at the time of our analysis. The reference refers instead to the study conducted by Kaise and Fukui on the possibility of metabolization of this element by sharks (through the liver, in fact) that may suggest that the As does not undergo biomagnification. However, biomagnification and bioaccumulation are two distinct processes. However, to avoid confusion for readers, we removed that sentence.
Point 9. So, please completely revise the presentation and discussion of your results. It is good to compare males and females, and the different locations. But this does not mean that there always has to be a “higher” or “lower”. Sometimes, there just is no difference. But this is also a valid finding. When comparing the locations with each other, it could be worth to also state how big the found differences are, so if one location should be regarded as polluted (and if so, then where does this pollution come from?), or if the values are all in an acceptable range. From what I could see from the data so far, it seems that the PW and GW values are always very similar to each other, and often lower than the others. Why? Could this have to do with their position, which seems a bit further away from coasts? I believe that some actually relevant, small discussions can be written from this dataset, to make the reader more interested in the study.
Response 9: We hope that the changes made, even following the revisions of other reviewers, will make the discussions more linear and understandable now. As for the comment on the two sites from the GSA 16 (PW and GW), it was already present from line 333 to line 341 but had an error (ST instead of GW) that we have now corrected thanks to your comment.
Point 10. To replace the rather irrelevant figures and tables (as mentioned above), I would love to see actual concentrations in a table. For example, Supplementary Table S3 could be moved to the main manuscript (+standard deviations). Further, the results for males vs. females could be presented as bar charts (+standard deviations) in the main manuscript.
Response 10: That was edited accordingly.
Point 11. Quality control:
I highly appreciate that a reference material is mentioned. It would be important that other information on quality assurance/quality control is given. For example: Were blanks measured? Drift controlled? What kind of calibration was done? How was the ICPMS operated (e.g. selected m/z values and gas modes, sensitivity)?
Response 11: We have added some of the information requested in the manuscript, with the hope that the description may be more comprehensive now.
Point 12: Sample preparation:
Why were the samples evaporated after digestion? What is meant by “glass matrasses”? How did you ensure that there was no significant contribution of the glass containers to the element concentrations? (Normal glass is very “dirty” when it comes to certain trace elements!) Why was “stirring” of the diluted samples necessary, and what was used for that purpose?
Response 12: We answered the questions by entering the missing information in the manuscript. As for the doubt about the use of glass, we specify that the permanence of the solution inside the glass container was minimal, just the time needed to volume and shake the solution, after this short phase the solution was immediately transferred into plastic containers washed with ultrapure water. Moreover, we are sure that there was no contamination by the material because, as now explained also in the manuscript, the CRM was prepared and treated with the same protocol as muscle tissue samples and - comparing the values obtained and certified values - no contamination was found.
Point 13. Official limits & co.: In L 290f, it is stated that the WHO classifies As as potentially essential. If such controversial claims are maid, it is crucial to give the appropriate references. Further, in L326f, the As results are compared to a “maximum limit of 15 mg/Kg”, which simply does not exist for foodstuff. Probably, the one cited is for feed, but this has no relevance here.
Response 13: We thank the reviewer for this remark and as we agree with his suggestions, we have completely removed this part from the text.
Point 14. References: For a rather short paper, there are a lot of references (81!). I suggest reducing the number of references, even though I appreciate the thoroughness. Several times, the DOI is not correct (e.g.: ref. 21 or 26, but also others). In other references, information is missing, such as the publication year (e.g. ref 7 or 12, but also others). Please check all references for correct and complete information.
Response 14: We removed the unnecessary references and checked the bibliography as required. Thas was edited as required.
------------------------------
Smaller comments:
Point 15. Title: Remove “Rafinesque, 1810” to make the title more concise and easier to understand. Such detailed information on the investigated organism is important, but belongs to the introduction and the Materials & Methods sections.
Response 15. This was edited accordingly.
Point 16. L 81: Change “(Martin et al., 1989)” to a proper reference.
Response 16: This was edited accordingly.
Point 17. Materials and methods: The description of the sampling areas are confusing, because only the “larger” areas are specifically mentioned in the main text, and the individual 7 points only in Figure 1 and Table 1.
Response 17: This was edited accordingly.
Point 18. Table 1: Make sure that the text in the first row has enough space between the different columns.
Response 18: This was edited accordingly.
Point 19. Table 1: Since the sharks probably swim a bit up and down all the time, I don’t think it makes sense to give depths up to 0.5 m precision.
Response 19: This was edited accordingly.
Point 20. L 141: the “r” in “room temperature” should be lower case.
Response 20: This was edited accordingly.
Point 21. L 151: Change “[…]for17[…]” to “[…]for 17[…]”
Response 21: This was edited accordingly.
Point 22. L 155: Change “[…] the inductively coupled […]” to “[…] an inductively coupled […]”
Response 22: This was edited accordingly.
Point 23. L 159: The two references explain exactly the same procedure. Remove one of them.
Response 23. This was edited accordingly.
Point 24. L 222-224: This is the original text of the template. Remove.
Response 24. This was edited accordingly.
Point 25. L 279: add a closing “)” after the references.
Response 25: This was edited accordingly.
Point 26. Supplementary: Make sure that the table captions are on the same page as the actual tables.
Response 26: This was edited accordingly.
Round 2
Reviewer 3 Report
Thank you for your thorough response and for taking most of my suggestions into consideration. I know that it was a lot. I believe that the paper is much better now. For a few figures, there was the question whether you should remove them or not. You can leave them in, if you want. I would just make some small changes to them, as indicated below.
Small comments:
L 79: Add a space between “may” and “reach”.
L 79: I believe the correct term is “particulate matter”, and not “particulated matter”.
L 175-180: Explanation of DRC; this should be general knowledge, but it doesn’t really hurt the manuscript, so you might leave it in as well. However, I doubt that argon was used as reaction gas. Usually, something like hydrogen or oxygen is used. This would indeed be the most interesting information in this context.
L 185: Add as space between “dried” and “in”.
L 195: Thank you for the clarification about the reason for drying down the digests. As a suggestion for the future, just to think about: Most ICP-MS labs I know are not doing this (= they are not drying down the digests.) Instead, they are just diluting acid digests to 1-20% acid content, routinely. And the instruments are fine. You can run them for a very long time on diluted acid (e.g. 10%) and they will still be fine (except some consumables like tubings, but they should be replaced every now and then anyways). And actually, several elements need an acid environment to be stable in solution.
L 213: Delete “part of the”, and add a “)” after “(CRM”.
L 266: I believe you wanted to say “divided into” (or “separated into”) instead of “divided by”.
L 273 (Figure 3): If you want to keep this figure, this is ok. I just ask you to use less significant digits. So, just write 35, 6, 29, 35, 18 and 12 %.
L 283: Add a space between “differed” and “in”.
L 292: Add a space between “element” and “in”.
L 314 (Figure 4): I understand that you want to show the differences in elements and sites at the same time. I guess that it is not possible to use concentrations (eg mg/kg) instead of %, because the levels of the different elements are too different for that. However, it still makes no sense to say that these 7 sites sum up to 100%. There is no logical reason for this. So, my suggestion is: normalize all concentrations to the mean concentration of the element. This will still give you the same visualization, but the y-axis makes more sense.
L 347: What is “[NO_PRINTED_FORM]”?
Supplementary: Add SD values to Table S2 and S3.
Author Response
Thank you for your thorough response and for taking most of my suggestions into consideration. I know that it was a lot. I believe that the paper is much better now. For a few figures, there was the question whether you should remove them or not. You can leave them in, if you want. I would just make some small changes to them, as indicated below.
Thank you so much for your comment. We agree that the paper has now definitely improved thanks to your suggestions.
Small comments:
L 79: Add a space between “may” and “reach”.
We edited that accordingly.
L 79: I believe the correct term is “particulate matter”, and not “particulated matter”.
We edited that accordingly.
L 175-180: Explanation of DRC; this should be general knowledge, but it doesn’t really hurt the manuscript, so you might leave it in as well. However, I doubt that argon was used as reaction gas. Usually, something like hydrogen or oxygen is used. This would indeed be the most interesting information in this context.
I added the explanation on the DRC because in the previous revisions we were told that the operation of the instrument was unclear. As for the use of Argon, I made a mistake in the explanation of the same, which I have now corrected also in the manuscript: argon is used to generate the plasma of the torch, while in the reaction cell - to reduce interference - methane gas is used.
L 185: Add as space between “dried” and “in”.
We edited that accordingly.
L 195: Thank you for the clarification about the reason for drying down the digests. As a suggestion for the future, just to think about: Most ICP-MS labs I know are not doing this (= they are not drying down the digests.) Instead, they are just diluting acid digests to 1-20% acid content, routinely. And the instruments are fine. You can run them for a very long time on diluted acid (e.g. 10%) and they will still be fine (except some consumables like tubings, but they should be replaced every now and then anyways). And actually, several elements need an acid environment to be stable in solution.
Thank you for your suggestion.
L 213: Delete “part of the”, and add a “)” after “(CRM”.
We edited that accordingly.
L 266: I believe you wanted to say “divided into” (or “separated into”) instead of “divided by”.
We edited that accordingly.
L 273 (Figure 3): If you want to keep this figure, this is ok. I just ask you to use less significant digits. So, just write 35, 6, 29, 35, 18 and 12 %.
We edited that accordingly.
L 283: Add a space between “differed” and “in”.
We edited that accordingly.
L 292: Add a space between “element” and “in”.
We edited that accordingly.
L 314 (Figure 4): I understand that you want to show the differences in elements and sites at the same time. I guess that it is not possible to use concentrations (eg mg/kg) instead of %, because the levels of the different elements are too different for that. However, it still makes no sense to say that these 7 sites sum up to 100%. There is no logical reason for this. So, my suggestion is: normalize all concentrations to the mean concentration of the element. This will still give you the same visualization, but the y-axis makes more sense.
We have now amended the figure as per reviewer’s request. The new figure shows a ratio between the concentration of each TE per site and the overall mean concentration of the TE. Now the y-axis shows the number of location (7) and each staked bar the individual contribution of site.
L 347: What is “[NO_PRINTED_FORM]”?
It was an automatically generated text during bibliography corrections. We corrected it throughout the manuscript.
Supplementary: Add SD values to Table S2 and S3.
We have replaced tables S2 and S3 with Table 3 in the main manuscript (which contains the SD). Therefore, supplementary materials no longer exist.